# Minocycline decreases CCR2-positive monocytes in the retina and ameliorates photoreceptor degeneration in a mouse model of retinitis pigmentosa

Ryo Terauchi[1,2], Hideo Kohno [1]*, Sumiko Watanabe[2], Saburo Saito[3], Akira Watanabe[1], Tadashi Nakano[1]

1 Department of Ophthalmology, The Jikei University School of Medicine, Tokyo, Japan, 2 Division of Molecular and Developmental Biology, The Institute of Medical Science, The University of Tokyo, Tokyo, Japan, 3 Department of Molecular Immunology, The Jikei University School of Medicine, Tokyo, Japan

* kohnohideo@jikei.ac.jp

## Abstract

Retinal inflammation accelerates photoreceptor cell death caused by retinal degeneration. Minocycline, a semisynthetic broad-spectrum tetracycline antibiotic, has been previously reported to rescue photoreceptor cell death in retinal degeneration. We examined the effect of minocycline on retinal photoreceptor degeneration using *c-mer proto-oncogene tyrosine kinase* (*Mertk*)$^{-/-}$*Cx3cr1*$^{GFP/+}$*Ccr2*$^{RFP/+}$ mice, which enabled the observation of CX3CR1-green fluorescent protein (GFP)- and CCR2-red fluorescent protein (RFP)-positive macrophages by fluorescence. Retinas of *Mertk*$^{-/-}$*Cx3cr1*$^{GFP/+}$*Ccr2*$^{RFP/+}$ mice showed photoreceptor degeneration and accumulation of GFP- and RFP-positive macrophages in the outer retina and subretinal space at 6 weeks of age. *Mertk*$^{-/-}$*Cx3cr1*$^{GFP/+}$*Ccr2*$^{RFP/+}$ mice were intraperitoneally administered minocycline. The number of CCR2-RFP positive cells significantly decreased after minocycline treatment. Furthermore, minocycline administration resulted in partial reversal of the thinning of the outer nuclear layer and decreased the number of apoptotic cells, as assessed by the TUNEL assay, in *Mertk*$^{-/-}$*Cx3cr1*$^{GFP/+}$*Ccr2*$^{RFP/+}$ mice. In conclusion, we found that minocycline ameliorated photoreceptor cell death in an inherited photoreceptor degeneration model due to *Mertk* gene deficiency and has an inhibitory effect on CCR2 positive macrophages, which is likely to be a neuroprotective mechanism of minocycline.

## Introduction

Inflammation in the central nervous system, as well as the retina, is considered a complicating factor in degenerative diseases [1–5]. Moreover, retinal inflammation is considered to accelerate photoreceptor cell death (PCD) in retinal degeneration (RD), including age-related macular degeneration and retinitis pigmentosa [6]. Hence, the management of inflammation is

**Funding:** Grant information: JSPS KAKENHI Grant-in-Aid for Young Scientists (Start-up) Grant Number 25893253 (for HK), Grant-in-Aid for Young Scientists (B) Grant Number 15K20288 (for HK) and Grant-in-Aid (C) for Scientific Research Grant Number 19K09960 (for HK).

**Competing interests:** The authors have declared that no competing interests exist.

pivotal and presumably beneficial for patients with RD. The elucidation of inflammatory mechanisms for the management of RD is the major research focus.

Minocycline, a semisynthetic broad-spectrum tetracycline antibiotic, has anti-inflammatory properties [7]. Several studies, including our study, have demonstrated that minocycline can ameliorate PCD in RD [8–10]. However, the mechanism of PCD rescue by minocycline remains largely unknown. Two potential mechanisms have been suggested; the anti-apoptotic mechanism and anti-inflammatory mechanism [11]. The innate immune system, which exerts a rapid non-specific response to an antigen, has been implicated in the development of RD, including human age-related macular degeneration and retinitis pigmentosa [12]. In healthy retinas, microglia or guardians of the retina located in the outer and inner plexiform layers maintain retinal homeostasis [12]. However, in RD, microglia are activated and the activated microglia migrate to the outer retina and subretinal space, the space between the outer segments of photoreceptors and retinal pigment epithelium (RPE). Minocycline inhibits both microglial activation and migration and hence, microglial suppression by minocycline is considered the major mechanism of PCD rescue [11]. However, minocycline is not a specific drug. Furthermore, other protagonists of retinal inflammation (i.e., bone marrow-derived macrophages) invade the outer retina [10, 13, 14]. Therefore, delineating the cause and mechanism of rescue of photoreceptor cells in the degenerative stage by minocycline is important.

Recently, we generated *c-mer proto-oncogene tyrosine kinase* $(Mertk)^{-/-}Cx3cr1^{GFP/+}Ccr2^{RFP/+}$ mice. This enabled the observation of CX3CR1-green fluorescent protein (GFP)- and CCR2-red fluorescent protein (RFP)-positive cells in inherited RD without light damage or requirement of any non-physiological procedures such as doxycycline administration (widely used for tetracycline-controlled transcriptional activation) [15]. Before RD occurs, only *Cx3cr1* expression is observed, which corresponds to resting microglia [16]. In progressive RD, *Ccr2* expression is markedly increased [15].

In this study, we found that minocycline administration to $Mertk^{-/-}Cx3cr1^{GFP/+}Ccr2^{RFP/+}$ mice not only ameliorated PCD but also reduced the number of CCR2-RFP-positive cells in the outer retina and subretinal space. These results indicate that *Ccr2* suppression is one of the mechanisms of photoreceptor protection by minocycline.

## Materials and methods

### Animals

$Mertk^{-/-}Cx3cr1^{GFP/+}Ccr2^{RFP/+}$ mice were generated as previously described [15]. *Mertk* genotyping was performed using the following primers: wild type, forward 5′-GCTTTAGCCTCCCCAGTAGC-3′ and reverse 5′-GGTCACATGCAAAGCAAATG-3′; mutant, forward 5′-CGTGGAGAAGGTAGTCGTACATCT-3′ and reverse 5′-TTTGCCAAGTTCTAATTCCATC-3′. *Cx3Cr1* genotyping was performed using the following primers: wild-type, forward 5′-TCCACGTTCGGTCTGGTGGG-3′ and reverse 5′-GGTTCCTAGTGGAGCTAGGG-3′; *Cx3cr1* mutant, forward 5′-GATCACTCTCGGCATGGACG-3′ and reverse 5′-GGTTCCTAGTGGAGCTAGGG-3′. *Ccr2* genotyping was performed using the following primers: common, forward 5′-TAAACCTGGTCACCACATGC-3′; wild-type, reverse 5′-GGAGTAGAGTGGAGGCAGGA-3′; *Ccr2* mutant, reverse 5′-CTTGATGACGTCCTCGGAG-3′.

Equal numbers of male and female mice were used in this study. All mice were housed in the animal facility at the Jikei University School of Medicine and were maintained under a 12-h light (~10 lx)/dark cycle. All animal procedures and experiments were approved by the Jikei University School of Medicine Animal Care Committee and conformed to the recommendations of both the American Veterinary Medical Association Panel on Euthanasia and

the Association for Research in Vision and Ophthalmology Statement for the Use of Animals in Ophthalmic and Vision Research.

## Minocycline administration

Minocycline was purchased commercially from Sigma-Aldrich (St. Louis, MO, USA) and was dissolved in phosphate-buffered saline. Thereafter, minocycline was intraperitoneally administered once daily to $Mertk^{-/-}Cx3cr1^{GFP/+}Ccr2^{RFP/+}$ mice (age: 4–6 weeks). The dose of minocycline was either 50 or 100 mg/kg. Phosphate-buffered saline was administered to the control group.

## Flat-mount retina and RPE preparation

All procedures for retinal and RPE flat-mounts were performed as previously described [10]. Images of flat-mounts were captured using a confocal microscope (LSM; Carl Zeiss, Thornwood, NY, USA). Regarding retinal flat-mounts, images of the entire retina were captured at 5 μm intervals and all images were projected in one slice. Regarding RPE flat-mounts, images of the entire visible RPE were captured at 3 μm intervals and projected in one slice.

## Histological analysis

All retinal sections were prepared as previously described [10, 17]. The numbers of CX3CR1-GFP- and CCR2-RFP-positive cells were counted using ImageJ (National Institutes of Health, Bethesda, MD, USA). To detect RFP, anti-RFP mouse antibody (MBL, M165-3) was used as the primary antibody, and the signals were visualized using a horseradish peroxidase-tagged anti-mouse IgG antibody (GE Healthcare, NA9310V) and a peroxidase-diaminobenzidine kit (Nacalai Tesque, Kyoto, Japan). Immunohistochemical images were captured using a confocal microscope (LSM 880; Carl Zeiss, Thornwood, NY, USA).

## Apoptosis assays

Retinas were frozen, sectioned, and subjected to the TUNEL assay using an in situ apoptosis detection kit (MK500, Takara Bio, Shiga, Japan).

## Flow cytometry analysis

The effect of minocycline on circulating monocytes was examined by flow cytometry analysis. Peripheral blood was collected from 16-week-old $Mertk^{-/-}Cx3cr1^{GFP/+}Ccr2^{RFP/+}$ mice and was incubated with anti-Ly6C conjugated APC-Cy7 antibody (BioLegend, 128025) on ice for 30 min. Thereafter, cells were washed and stained with propidium iodide to exclude dead cells and analyzed using a BD FACSAria III Cell Sorter (BD Biosciences, San Jose, CA, USA). Data were analyzed using FlowJo software version 10.7.1 (Tree Star, Ashland, OR, USA).

## Data analysis

Continuous variables are presented as mean ± standard deviation. The Steel–Dwass test was performed for non-parametric multiple comparisons between the groups. All statistical analyses were performed using the statistical program R (version 4.0.3; R Foundation for Statistical Computing). Statistical significance was set at $P < 0.05$.

## Results

### Characterization of the *Mertk*$^{-/-}$*Cx3cr1*$^{GFP/+}$*Ccr2*$^{RFP/+}$ mouse retina

In *Mertk*$^{-/-}$*Cx3cr1*$^{GFP/+}$*Ccr2*$^{RFP/+}$ mice, the descriptions "4-" and "6-week-old" correspond to the retinal non-degenerative and RD ongoing stage, respectively [15]. Representative CX3CR1-GFP single-positive cells observed in retinal flat-mounts of 4-week-old *Mertk*$^{-/-}$*Cx3cr1*$^{GFP/+}$*Ccr2*$^{RFP/+}$ mice and CX3CR1/CCR2 double-positive cells observed in RPE flat-mounts of 6-week-old *Mertk*$^{-/-}$*Cx3cr1*$^{GFP/+}$*Ccr2*$^{RFP/+}$ mice are presented in Fig 1A and 1B. Time series vertical sections of *Mertk*$^{-/-}$*Cx3cr1*$^{GFP/+}$*Ccr2*$^{RFP/+}$ mice are presented in Fig 1C–1E. In 4-week-old mice, only CX3CR1-GFP single-positive cells were visible in the inner retina (Fig 1C). Neither CX3CR1-GFP- nor CCR2-RFP-positive cells were observed in the outer retina and subretinal space. In 3-month-old mice, RD, represented by thinning of the outer nuclear layer (ONL), was observed (Fig 1D). The number of ONL nuclei decreased from approximately 12 at 4 weeks to 1–4 at 3 months. Abundant CX3CR1-GFP- and CCR2-RFP-positive cells were observed in the ONL and subretinal space (Fig 1D). Some cells were CX3CR1 and CCR2 double-positive. In 1.5-year-old mice, almost all nuclei in the ONL had disappeared, indicating severe RD (Fig 1E). The frequency of CX3CR1-GFP- and CCR2-RFP-positive cells was found to be lower compared to the ongoing degeneration stage (e.g., from 6 weeks to 3 months).

### Minocycline administration reduced the number of CCR2-positive cells in neural retina

The numbers of CCR2-RFP-positive cells and CX3CR1-GFP and CCR2-RFP double-positive cells were reduced in the 50 and 100 mg/kg minocycline-treated (Mino50 and Mino100) groups than in the control group (Fig 2E and 2F). In retinal flat-mounts, retinal layer boundaries are difficult to observe. However, many CCR2-RFP-positive cells were observed in the outer plexiform layer and ONL in the 3D images of the control group (Fig 2B). In the 3D image of the Mino100 group, CCR2-RFP was hardly detected (Fig 2C). On the contrary, the number of CX3CR1-GFP-positive cells was not affected by minocycline administration (Fig 2D).

### Minocycline administration reduced the number of CCR2 positive cells in the subretinal space

RPE flat-mounts were prepared to observe the apical side of the RPE, corresponding to the subretinal space (Fig 3) [10, 15, 18]. In the RPE flat-mounts from 4-week-old *Mertk*$^{-/-}$*Cx3cr1*$^{GFP/+}$*Ccr2*$^{RFP/+}$ and *Mertk*$^{+/+}$*Cx3cr1*$^{GFP/+}$*Ccr2*$^{RFP/+}$ mice that did not show RD, neither CX3CR1-GFP positive cells nor CCR2-RFP positive cells were observed [15]. Abundant CX3CR1-GFP positive cells were observed in the subretinal space in the control, Mino50, and Mino100 groups (Fig 3A). The number of CCR2-RFP positive cells decreased in the Mino50 and Mino100 groups compared with the control group (Fig 3E and 3F); however, the number of CX3CR1-GFP positive cells did not change in all the groups (Fig 3D), indicating that minocycline administration probably did not affect the migration of CX3CR1-GFP positive cells to the subretinal space but suppressed the migration of CCR2-positive cells.

Thereafter, we confirmed that the fluorescent signals in the subretinal region were not caused by auto-fluorescence by a colorimetric assay. Immunohistochemistry was performed on the frozen sections of retinas of 3-month-old *Mertk*$^{-/-}$*Cx3cr1*$^{GFP/+}$*Ccr2*$^{RFP/+}$ mice by the

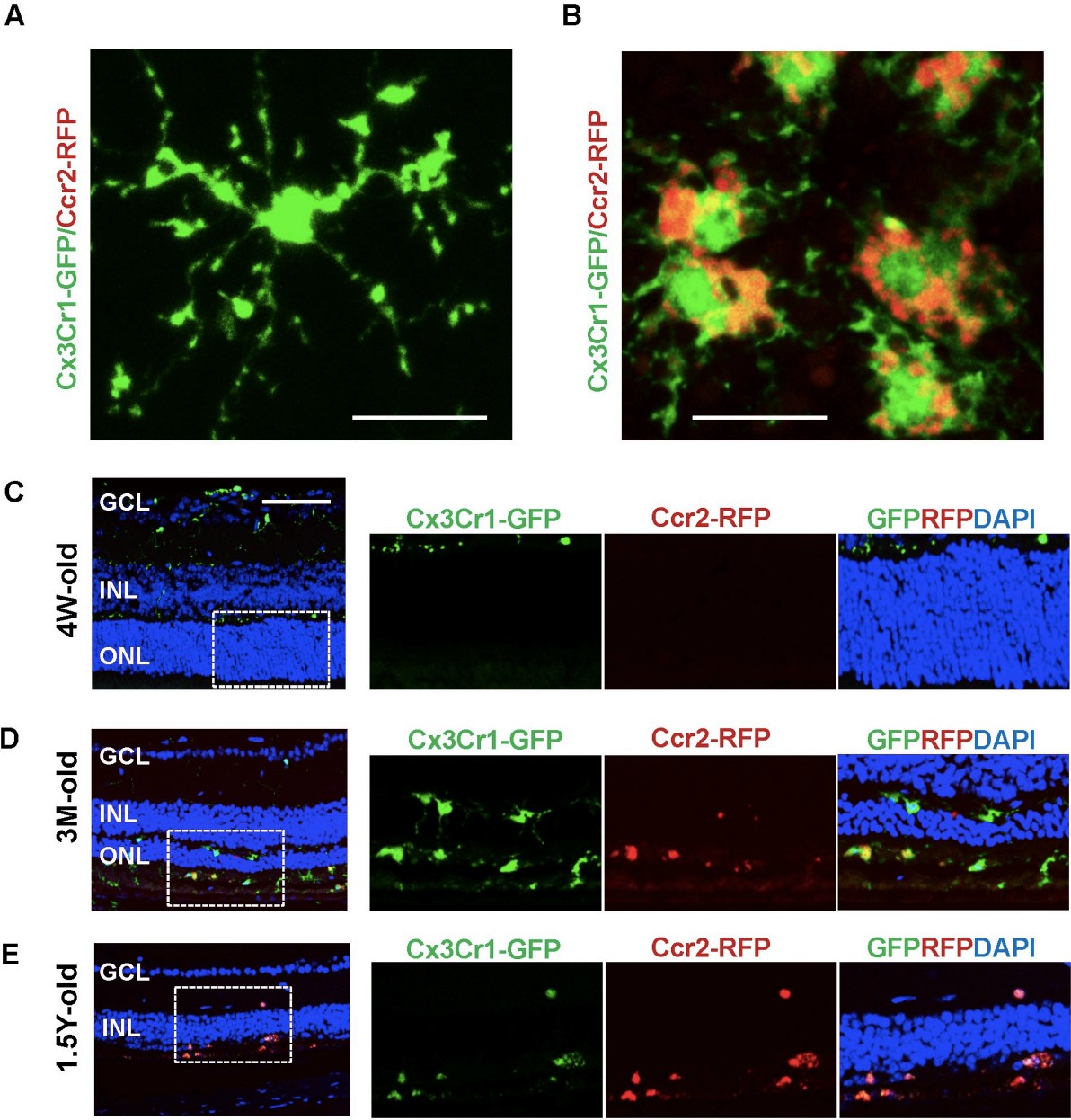

**Fig 1. Characterization of *Mertk<sup>−/−</sup>Cx3cr1<sup>GFP/+</sup>Ccr2<sup>RFP/+</sup>* mice.** (A) Magnified CX3CR1 single-positive cells observed in retinal flat-mounts from 4-week-old *Mertk<sup>−/−</sup>Cx3cr1<sup>GFP/+</sup>Ccr2<sup>RFP/+</sup>* mice and (B) CX3CR1-GFP and CCR2-RFP double-positive cells in RPE flat-mounts from 6-week-old *Mertk<sup>−/−</sup>Cx3cr1<sup>GFP/+</sup>Ccr2<sup>RFP/+</sup>* mice (scale bar = 20 μm). (C-E) Vertical sections from 4-week-, 3-month-, and 1.5-year-old *Mertk<sup>-/-</sup>Cx3cr1<sup>GFP/+</sup>Ccr2<sup>RFP/+</sup>* mice (scale bar = 50 μm). Inserts are the magnified images. GCL, ganglion cell layer; INL, inner nuclear layer; ONL, outer nuclear layer; RPE, retinal pigmented epithelium.

peroxidase-diaminobenzidine method. Diaminobenzidine signals were observed in the outer retina (Fig 4A), which was consistent with the fluorescence observation (Fig 1D). Diaminobenzidine staining revealed that minocycline treatment decreased the number of CCR2-RFP positive cells (Fig 4B and 4C).

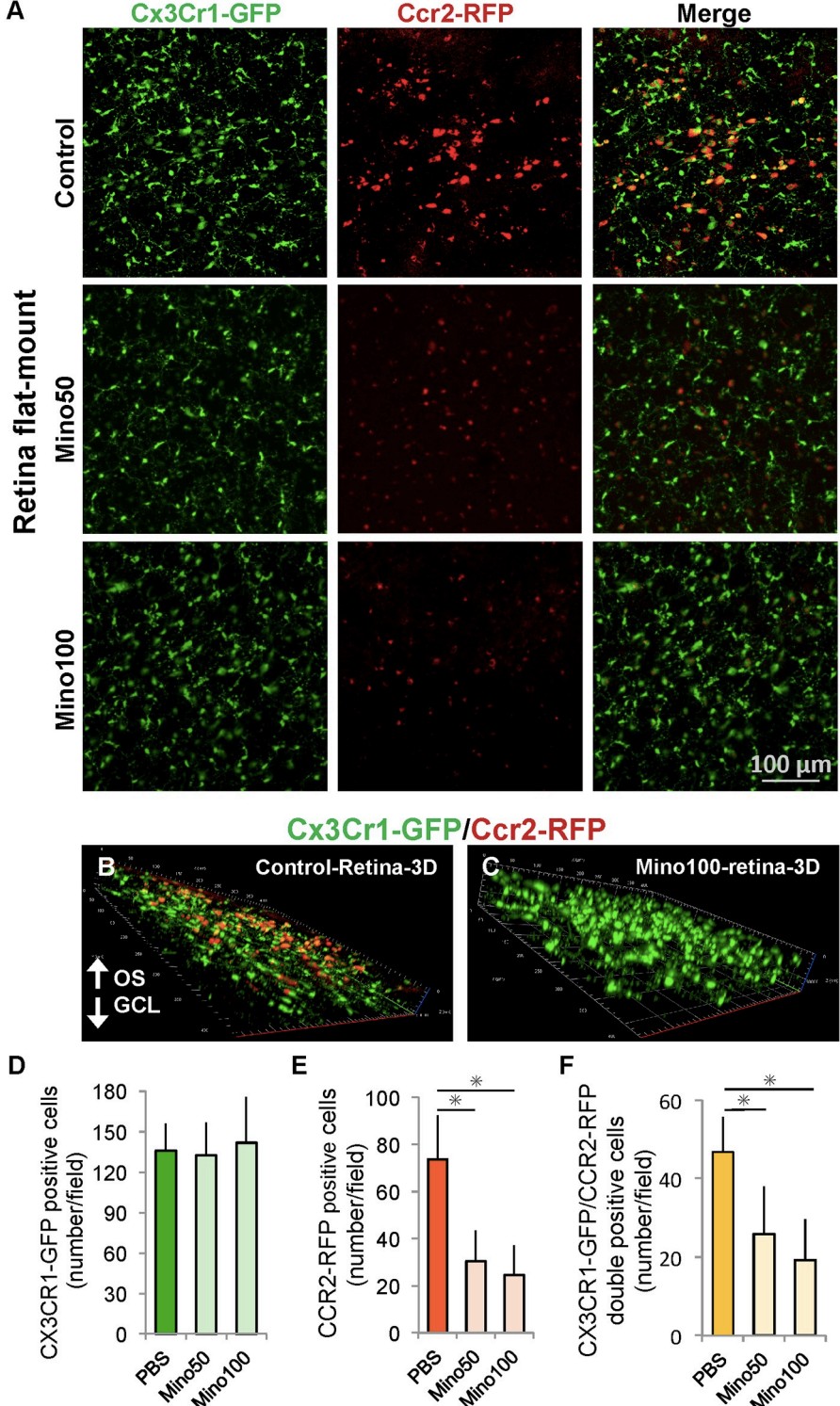

**Fig 2. Minocycline administration reduced CCR2-positive cells in retinal flat-mounts.** An intraperitoneal injection of minocycline was administered daily for 14 days to $Mertk^{-/-}Cx3cr1^{GFP/+}Ccr2^{RFP/+}$ mice (age: 4–6 weeks). The minocycline group was divided into two subgroups: the Mino50 (50 mg/kg) and Mino100 (100 mg/kg) groups. Phosphate-buffered saline (PBS) was administered to the control group. (A) The retinal flat-mounts of each group were prepared after treatment at 6 weeks of age and observed by laser confocal microscopy. (B) and (C) 3D images of the control and Mino100 groups. The numbers of (F) CX3CR1-GFP positive (GFP+RFP− and GFP+RFP+) cells, (G)

CCR2-RFP positive (GFP-RFP+ and GFP+RFP+) cells, and (H) CX3CR1-GFP and CCR2-RFP double-positive (GFP +RFP+) cells in each group (n ≥ 5 per group). * indicates $P < 0.05$. OS, outer segment. GFP, green fluorescent protein; RFP, red fluorescent protein.

## Amelioration of PCD by minocycline administration

Finally, we evaluated the therapeutic effects of minocycline in $Mertk^{-/-}Cx3cr1^{GFP/+}Ccr2^{RFP/+}$ mice. Previously, we reported PCD amelioration by minocycline administration in a light-induced acute RD mouse model ($Abca4^{-/-}Rdh8^{-/-}$ mice) [10]. However, the therapeutic effect of minocycline in inherited RD due to $Mertk$ gene deficiency was unknown. First, minocycline was administered to 4-week-old $Mertk^{-/-}Cx3cr1^{GFP/+}Ccr2^{RFP/+}$ mice for 2 weeks. However, the severity of PCD did not differ between the minocycline-treated and control mice because PCD was relatively mild at the age of 6 weeks in $Mertk^{-/-}Cx3cr1^{GFP/+}Ccr2^{RFP/+}$ mice. Thereafter, minocycline (50 mg/kg) or PBS was administered daily for 2 weeks from the age of 6 weeks to $Mertk^{-/-}Cx3cr1^{GFP/+}Ccr2^{RFP/+}$ mice (Fig 5). Minocycline-treated $Mertk^{-/-}Cx3cr1^{GFP/+}Ccr2^{RFP/+}$ mouse retinas showed fewer CCR2-RFP positive cells in the subretinal region compared with control mouse retinas (Fig 5A). The thickness of the ONL was retained in the central region but not in the peripheral region (Fig 5B and 5C), indicating amelioration of PCD by minocycline. In addition, minocycline significantly suppressed the proportion of TUNEL-positive photoreceptor cells (Fig 5D and 5E). The ONL thickness in 4-week-old $Mertk^{-/-}Cx3cr1^{GFP/+}Ccr2^{RFP/+}$ and wild type (B6) mice are shown as negative controls in S1 Fig.

## Long-term minocycline administration

To examine whether minocycline has protective effects against photoreceptor degeneration in the later stage in the $Mertk$ knockout model, $Mertk^{-/-}Cx3cr1^{GFP/+}Ccr2^{RFP/+}$ mice were treated with minocycline from 8-week-old to 16-week-old. In the control group, the number of CCR2-RFP positive cells accumulated in the subretinal space (Fig 6A). On the other hand, fewer CCR2-RFP positive cells were observed in retinas of minocycline (50 and 100 mg/kg)-treated mice (Fig 6A and 6B). The ONL thickness in minocycline-treated retina was retained, especially in the central region of the retina (Fig 6C), and the numbers of cells in the ONL of the Mino50 and Mino100 groups were significantly higher than that of the control group (Fig 6D).

To examine whether minocycline treatment affects circulating monocytes, we examined Ly6C positive monocytes in the peripheral blood of control, Mino50, and Mino100 treated $Mertk^{-/-}Cx3cr1^{GFP/+}Ccr2^{RFP/+}$ mice (S2A Fig). We found no significant difference in the proportion of Ly6C positive cells among control and minocycline-treated mice (S1B Fig).

## Discussion

Minocycline is considered an anti-inflammatory and neuroprotective drug candidate for RD [8, 19]. In this study, we investigated the therapeutic effects of minocycline on RD induced by $Mertk$ knockout. These mice contained GFP and RFP in the $Cx3cr1$ and $Ccr2$ loci, respectively, allowing us to monitor and evaluate microglia and macrophages during degeneration. We found that minocycline administration suppressed the number of CCR2-RFP positive cells and partially protected photoreceptor degeneration.

In $Mertk$ knockout models, CCR2-RFP single and CCR2-RFP and Cx3cr1-GFP double-positive cells accumulated in the outer retina and subretinal space, which is consistent with the findings of our study [15]. CCR2 is regarded as an essential chemokine receptor for macrophage recruitment to inflammation sites [20] and is detectable in monocyte-derived

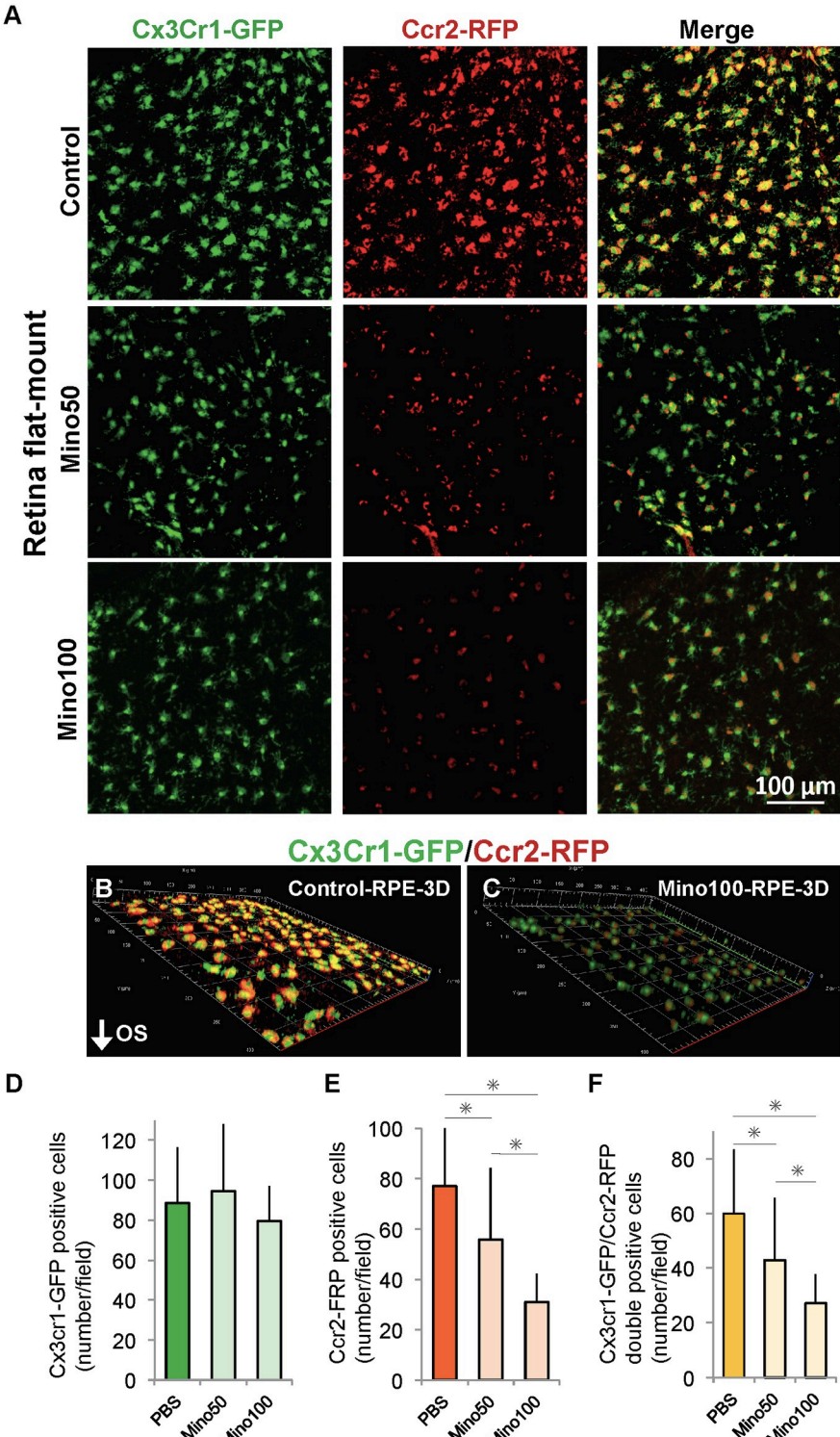

**Fig 3. Minocycline administration reduced CCR2-positive cells in RPE flat-mount.** An intraperitoneal injection of minocycline was administered daily for 14 days to $Mertk^{-/-}Cx3cr1^{GFP/+}Ccr2^{RFP/+}$ mice (age: 4–6 weeks). (A) RPE flat-mounts from control, Mino50, and Mino100 were prepared after treatment at 6 weeks of age. (B) and (C) 3D images of the control and Mino100. The numbers of (D) CX3CR1-GFP positive (GFP+RFP– and GFP+RFP+) cells, (E) CCR2-positive (GFP–RFP+ and GFP+RFP+) cells, and (F) CX3CR1-GFP and CCR2-RFP double-positive (GFP+RFP+) cells in each group (n ≥ 5 per group). * indicates $P < 0.05$. GFP, green fluorescent protein; RFP, red fluorescent protein.

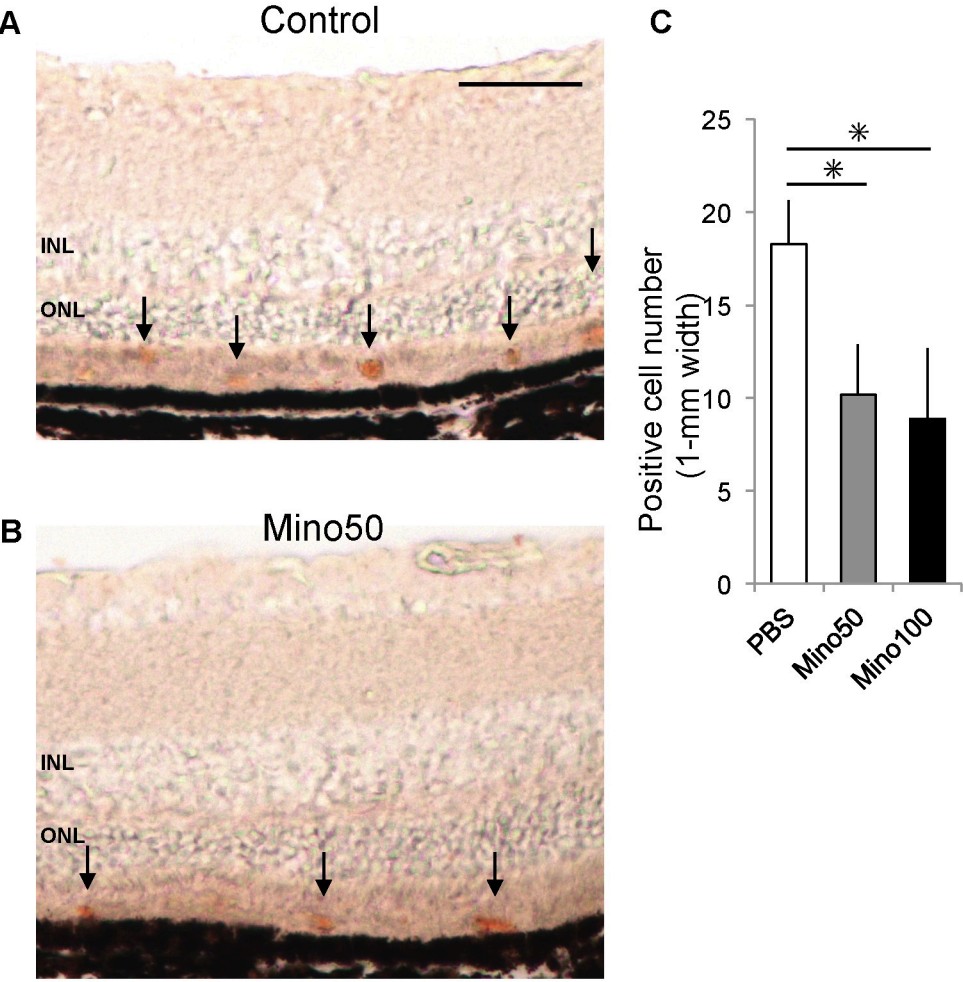

**Fig 4. Diaminobenzidine staining for CCR2-RFP positive cells.** (A) and (B) Frozen sections of retinas derived from 3-month-old control and minocycline (50 and 100 mg/kg)-treated $Mertk^{-/-}Cx3cr1^{GFP/+}Ccr2^{RFP/+}$ mice were immunostained with anti-RFP antibody. Signals were visualized using the peroxidase-diaminobenzidine method. Positive signals are indicated by black arrows (scale bar = 50 μm). The number of positive cells in 1-mm wide immunostained sections of control and minocycline (50 and 100 mg/kg)-treated retinas were counted (n ≥ 5 per group) (C) * indicates $P < 0.05$. RFP, red fluorescent protein.

macrophages but not in resident microglia [21]. Therefore, CCR2-RFP positive cells observed in the current study were considered macrophages. However, it should be noted that the localization of monocyte-derived macrophage in RD varies among degeneration models, that is, the retinal ischemia-reperfusion model showed the accumulation of macrophages mainly in the ganglion cell layer and inner plexiform layer [22]. Light damage induced only activated resident microglia, but not monocyte-derived macrophages, into the subretinal space [23]. These findings suggest that the effects of minocycline vary depending on the degeneration model.

Recently, it was reported that minocycline has an effect on the CCL2/CCR2 pathway and macrophage infiltration. In primary microglia (in vitro) and neuropathic pain rat models (in vivo), minocycline suppressed the upregulation of CCL2 and CCR2 [24]. In a mouse model of cerebellar hemorrhage, infiltration of monocytes and macrophages into the cerebellum was decreased by minocycline treatment [25]. In the current study, significant inhibition of the number of CCR2-RFP positive cells in both the retina and RPE was observed in flat-mounts.

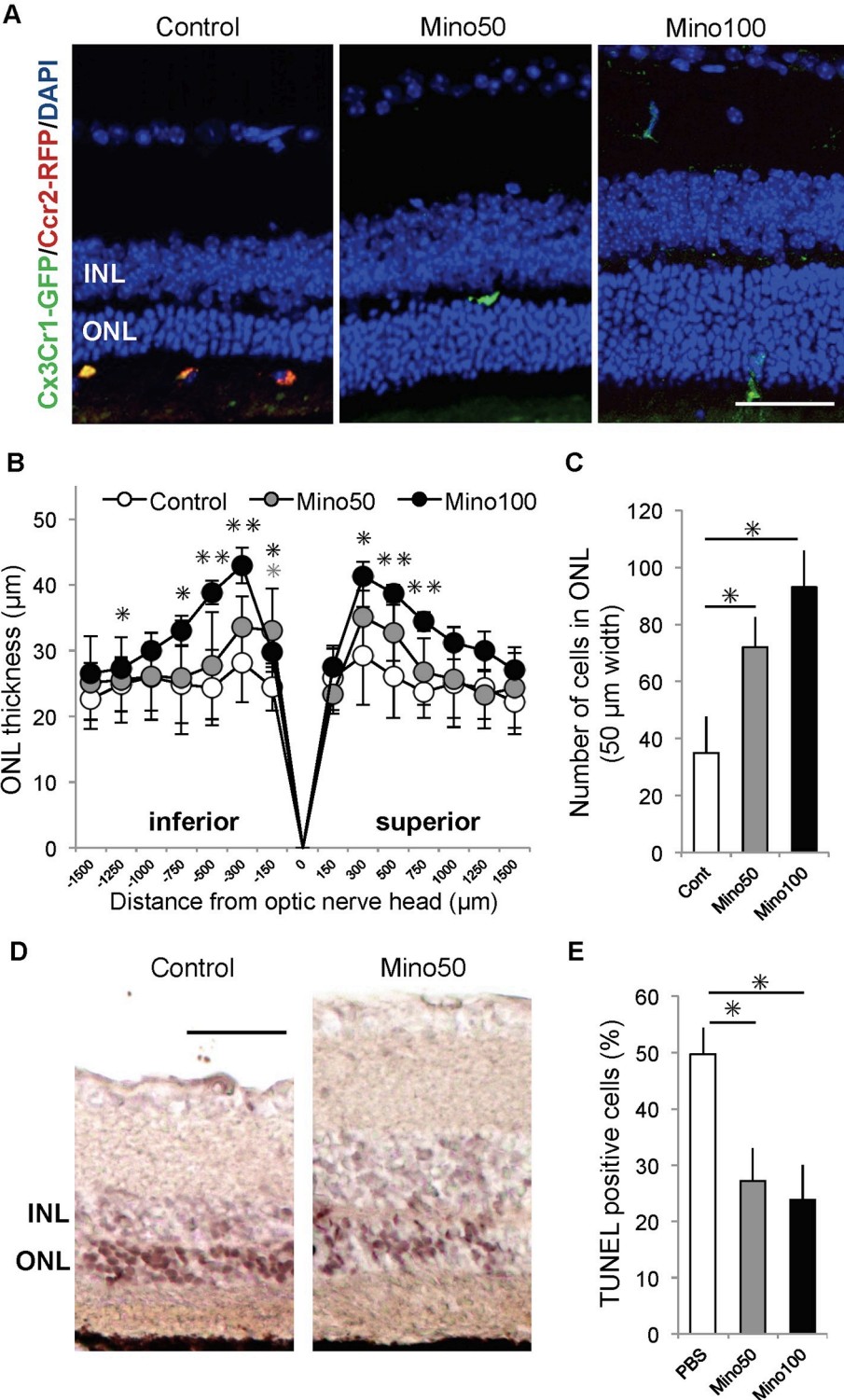

**Fig 5. Minocycline administration ameliorates photoreceptor cell death in *Mertk$^{-/-}$Cx3cr1$^{GFP/+}$Ccr2$^{RFP/+}$* mice.**
Minocycline (50 mg/kg [Mino50] and 100 mg/kg [Mino100]) or phosphate-buffered saline (PBS; control) was administered once daily to *Mertk$^{-/-}$Cx3cr1$^{GFP/+}$Ccr2$^{RFP/+}$* mice (age: 6–8 weeks). (A) Retinas were frozen and sectioned, and nuclei were visualized by DAPI staining. (B) Thickness and (C) the number of cells of the ONL were measured (n ≥ 5 per group). (D) Apoptotic cells in frozen sections of retinas of control and minocycline (50 mg/kg)-treated *Mertk$^{-/-}$Cx3cr1$^{GFP/+}$Ccr2$^{RFP/+}$* mice were examined by the TUNEL assay. (E) Proportion of TUNEL-positive

cells in the control, Mino50, and Mino100 groups are shown. (B), (C), and (E): * indicates $P < 0.05$. (B): ** indicates $P < 0.01$. Gray and black asterisks indicate $P$ values of control vs. Mino50 and control vs. Mino100, respectively. (A) and (D): Scale bar = 50 μm. DAPI, 4',6-diamidino-2-phenylindole; ONL, outer nuclear layer; TUNEL, TdT-mediated dUTP nick end labeling.

This finding is consistent with the findings of previous studies. However, to fully understand the neuroprotective mechanisms of minocycline, it should be clarified whether minocycline suppressed the CCL2-induced CCR2 signaling pathway and whether minocycline decreased the recruitment of CCR2-expressing monocytes or the CCR2 expression level in monocytes.

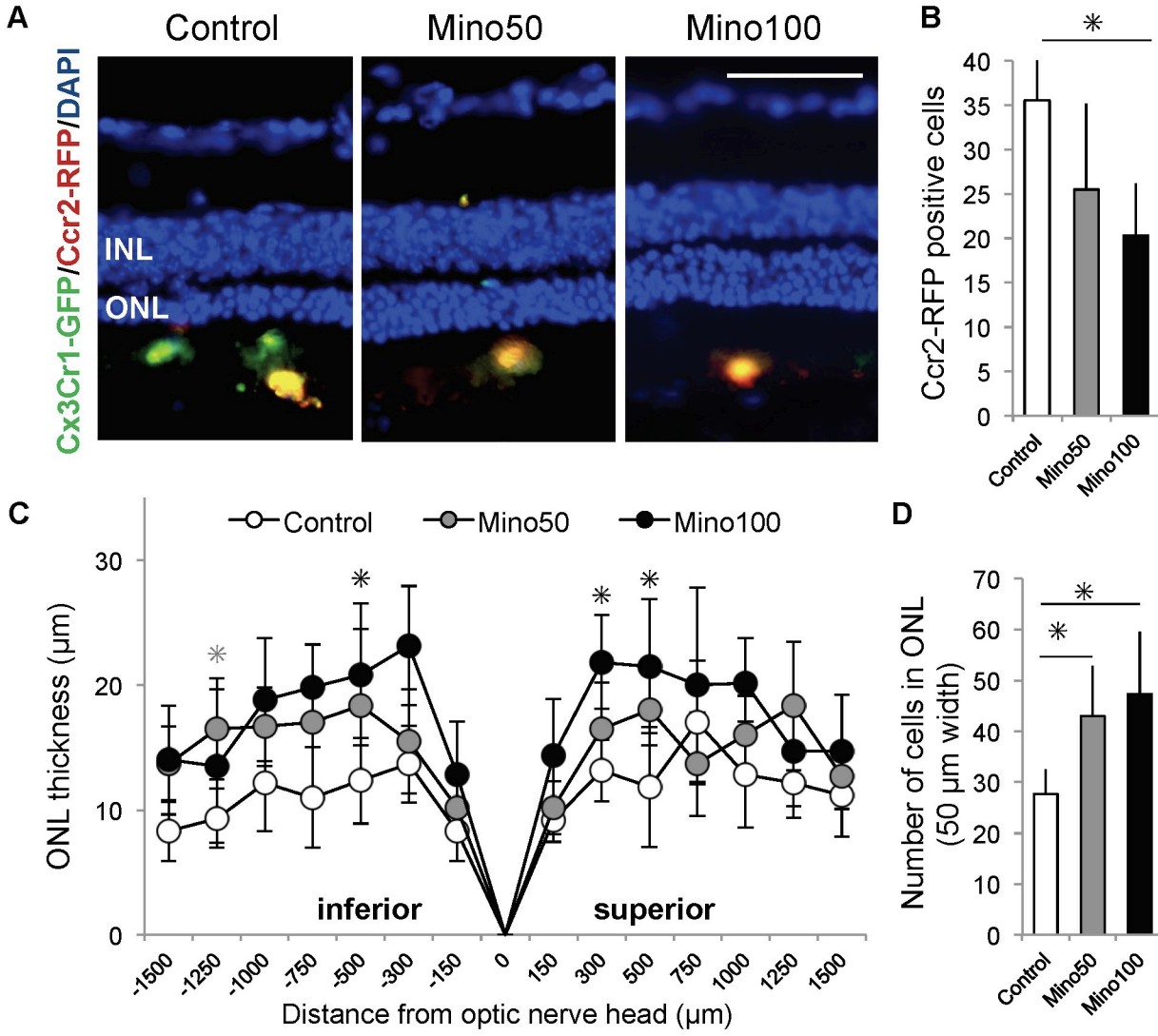

**Fig 6. Long-term minocycline administration ameliorates photoreceptor cell death in $Mertk^{-/-}Cx3cr1^{GFP/+}Ccr2^{RFP/+}$ mice.** Minocycline (50 mg/kg; Mino50), minocycline (100 mg/kg; Mino100), or phosphate-buffered saline (PBS; control) was administered once daily to $Mertk^{-/-}Cx3cr1^{GFP/+}Ccr2^{RFP/+}$ mice from 8 to 16 weeks of age. Retinas were frozen and sectioned, and nuclei were visualized by DAPI staining (A) (scale bar = 50 μm). The number of CCR2-RFP positive cells in the subretinal space was counted. Values are the average of five independent samples with standard deviations. Thickness (B) and the number of cells (C) of the ONL were measured (n ≥ 5 per group). * indicates $P < 0.05$. (C): the gray and black asterisks indicate $P$ values of control vs. Mino50 and control vs. Mino100, respectively. DAPI, 4',6-diamidino-2-phenylindole; ONL, outer nuclear layer; RFP, red fluorescent protein.

Further, whether the inhibition of the CCL2/CCR2 pathway in macrophages and monocyte infiltration suppress RD remains unclear. The anti-inflammatory and neuroprotective effects of CCL2/CCR2 suppression in RD have been reported [15, 26]; however, a study reported that the elimination of the CCL2/CCR2 pathway inhibited monocyte infiltration, but did not block neurodegeneration in a light injury mouse model [13]. This issue should be resolved to elucidate the neuroprotective effects of minocycline.

We found that minocycline had a dose-dependent neuroprotective effect on photoreceptor cells (Figs 5 and 6). However, it should be noted that no significant difference was noted in the peripheral ONL thickness between control and minocycline-treated retinas. These findings suggest that minocycline was not completely protective against photoreceptor cell degeneration, which is suggested by the finding that 20%–30% of photoreceptor cells were stained with TUNEL after minocycline treatment (Fig 5D and 5E).

Because macrophages infiltrating the retina are thought to subsequently reduce *Ccr2* expression [21], it is possible that the number of macrophages was misled by the altered expression level of *Ccr2*. To accurately verify the extent of monocyte infiltration inhibition by minocycline, an additional technique that specifically labels all retinal macrophages should be used to distinguish them from resident microglia.

In the current study, we investigated the neuroprotective effects of minocycline using a Mertk knockout mouse model. Consequently, minocycline ameliorated PCD in inherited RD due to *Mertk* gene deficiency and reduced the total number of CCR2-RFP positive monocyte-derived macrophages accumulating in the outer retina and subretinal space. The suppression of the chemokine receptor CCR2 in retinal macrophages might be one of the neuroprotective mechanisms of minocycline. Further studies are warranted to examine whether minocycline decreases the number of infiltrating monocytes or suppresses Ccr2 expression in monocytes.

## Supporting information

**S1 Fig. The outer nuclear layer thickness of 4-week-old *Mertk$^{-/-}$Cx3cr1$^{GFP/+}$Ccr2$^{RFP/+}$* and wild type mice.** The outer nuclear layer thickness of 4-week-old *Mertk$^{-/-}$Cx3cr1$^{GFP/+}$Ccr2$^{RFP/+}$* mice and wild type (B6) mice are presented as negative controls in Fig 5.
(TIFF)

**S2 Fig. The population of Ly6C positive circulating monocytes in minocycline- and control-treated *Mertk$^{-/-}$Cx3cr1$^{GFP/+}$Ccr2$^{RFP/+}$* mice.** (A) Ly6C positive monocytes were detected in control *Mertk$^{-/-}$Cx3cr1$^{GFP/+}$Ccr2$^{RFP/+}$* mice by flow cytometry analysis. (B) The proportion of Ly6C positive monocytes in the control, Mino50, and Mino100 groups (n = 3 per group).
(TIFF)

## Author Contributions

**Data curation:** Ryo Terauchi, Hideo Kohno, Sumiko Watanabe, Saburo Saito, Akira Watanabe.

**Investigation:** Ryo Terauchi, Hideo Kohno, Akira Watanabe.

**Project administration:** Hideo Kohno.

**Resources:** Hideo Kohno.

**Supervision:** Sumiko Watanabe, Saburo Saito, Akira Watanabe, Tadashi Nakano.

**Writing – original draft:** Ryo Terauchi, Hideo Kohno, Sumiko Watanabe.

**Writing – review & editing:** Ryo Terauchi, Hideo Kohno, Sumiko Watanabe.

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
