## [Decision Letter · Decision Letter 0]

5 Oct 2020

PONE-D-20-26780

Ccr2 suppression by minocycline in Cx3cr1/Ccr2-visualized inherited retinal degeneration

PLOS ONE

Dear Dr. Kohno,

Thank you for submitting your manuscript to PLOS ONE. After careful consideration, we feel that it has merit but does not fully meet PLOS ONE’s publication criteria as it currently stands. Therefore, we invite you to submit a revised version of the manuscript that addresses the points raised during the review process.

Neither of the expert reviewers felt that your results fully support the conclusions you have drawn from them.  In addition one of the reviewers indicated that your title and abstract were imprecise and confusing. Both reviewers provided detailed critiques that should help you in revising the paper.

We look forward to receiving your revised manuscript.

Kind regards,

Alfred S Lewin, Ph.D.

Academic Editor

PLOS ONE

Journal Requirements:

2. To comply with PLOS ONE submissions requirements, please provide methods of sacrifice in the Methods section of your manuscript.

Reviewers' comments:

Reviewer's Responses to Questions

**Comments to the Author**

1. Is the manuscript technically sound, and do the data support the conclusions?

Reviewer #1: Partly

Reviewer #2: Partly

2. Has the statistical analysis been performed appropriately and rigorously? 

Reviewer #1: I Don't Know

Reviewer #2: Yes

3. Have the authors made all data underlying the findings in their manuscript fully available?

Reviewer #1: Yes

Reviewer #2: Yes

4. Is the manuscript presented in an intelligible fashion and written in standard English?

Reviewer #1: Yes

Reviewer #2: Yes

5. Review Comments to the Author

Reviewer #1: This manuscript describes the effect of minocycline, an approved antibiotic that has also been investigated clinically in neurodegenerative diseases of the retina and CNS, on a mouse model of inherited retinal degeneration. The manuscript focuses on the population of innate immune cells in the outer retina during degeneration, correlating them to the amount of PR degeneration that occurs. The authors found that in this model, systemic minocycline resulted in a decreased prevalence of CCR2+ cells in the outer retina that correlated with a reduction in photoreceptor degeneration.

This work is relevant and topical in that (1) modulating neuroinflammatory drivers of inherited retinal degenerations is a therapeutic strategy under current investigation, and (2) the functional roles of innate immune cell types (microglia and monocytes) are unknown and a subject of recent scrutiny. The manuscript can be improved by establishing more clarity in the number of points in the results and discussion.

1. Title: The current title is ambiguous and can benefit from rephrasing. Both “Ccr2 suppression” and “Cx3cr1/Ccr2-visualized” terms are imprecise and hard to understand. I suggest stating the main conclusions in the title: “Minocycline administration diminishes CCR2-positive monocyte number in the retina and ameliorates photoreceptor degeneration in a mouse model of retinitis pigmentosa”.

2. Abstract: The terms here and elsewhere relating to the suppression of Ccr2 expression are imprecise. What the authors document here is a decrease in the prevalence of CCR2-RFP+ cells in the outer retina and RPE layers with minocycline. Whether this finding resulted from the decreased recruitment of CCR2-expressing monocytes versus the actual suppression of CCR2 expression in monocytes resulting in a decrease in RFP-fluorescence is not investigated. The authors may want to reconsider the use of terms that indicate that CCR2 expression is decreased in individual monocytes from minocycline’s action (like “CCR2 suppression” ). The authors should pay attention to the CCR2 vs. Ccr2 case nomenclature throughout the manuscript to make sure the correct term is used for gene and protein names. The abstract can probably be shortened with the omission of methodological detail.

3. Results: It is important for the authors to establish what is being referred to as Cx3cr1-positive, Ccr2-positive and Cx3cr1/Ccr2 dual-positive cells are indeed GFP+/RFP-, GFP-/RFP+, and GFP+/RFP+ cells respectively. I had been looking throughout the manuscript for a confirmatory statement of this but I was not able to find it. The confusion is increased in Fig 2 and 3 in which the numbers in A-4, B-4, and C-4 do not match corresponding numbers in F, G, and H; in F, G, and H; dual-positive cells constitute the smallest constituency of all 3 cellular groups for all 3 conditions but in the pie charts, they consistently outnumber CCR2-postive cells. Without harmonizing this confusion, it is not possible to fully understand the data.

4. Results: The ameliorative effect of minocycline on ONL thickness is actually quite minor and localized, leading one to wonder if it can be validated if more experimental numbers are accumulated. As such, the reference to this difference as “rescue” may be wording it too strongly. For smaller effects like this, more evidence is often desired to strengthen the conclusion, these include: (a) a dose dependent effect (i.e. more amelioration with higher doses of minocycline, like 100mg), (b) independent method of ascertaining ONL thickness (e.g. with in vivo OCT measurements), and (c) other measurements of PR death (e.g. TUNEL labeling of apoptotic cells, length of outer segments, counting cone density etc). If these are available, they can help strengthen the conclusion that minocycline confers protection in this model.

5. Discussion: The authors can focus more on interpreting the main findings of the manuscript, asking the questions of (a) whether minocycline inhibits infiltration of CCR2-expressing monocytes into the outer retina, or whether CCR2+ monocytes are converted into dual positive cells, to CX3CR1+, CCR2- cells, or even to CX3CR1-, CCR2- cells (which may render them undetectable in this assay) and (b) whether this effect of minocycline on CCR2-expressing monocytes is causally connected with the amelioration of PR degeneration, and if so, by what putative mechanisms. These discussion points follow directly from the main results and should constitute the main content of the discussion.

Reviewer #2: In this manuscript Terauchi et al investigate the effect of minocycline on CX3CR1+CCR2- microglial cells, and CX3CR1+CCR2+ and Cx3cr1-CCR2+ monocytes/macrophages. Using their previously published Mertk-/-CX3CR1GFP/+CCR2RFP/+ mice they demonstrate that minocycline inhibits the recruitment of CCR2+ cells and slightly reduces the observed degeneration. This is a potentially interesting study as it might clarify the question if and to what extent monocytes are recruited to the subretinal space in RP (at least in Mertk-/-) and help understand the anti-inflammatory effect of minocycline in neurogenerative diseases. There are several points that I think need strengthening:

Autofluorescence of subretinal mononuclear phagocytes: O’Koren claimed that the subretinal “niche” can not be infiltrated by monocyte derived cells but only by microglial cells. One incertainty in both studies is that they only use fluorescent markers to identify the cell types. Yet subretinal macrophages (whatever type, microglial or monocyte derived) are known to become very autofluorescent as they phagocyte visual pigment rich outer segments of the photoreceptors. In any of these studies it would therefore be important that other techniques are used to corroborate the results using non-fluorescent techniques, such as substrate based immunohistochemistry (e.g. Fast red, if there is an antibody that differentiates GFP and RFP that should be easy), or in this case a difference in CCR2 expression by RT-PCR (or other Monocyte markers Ly6C etc…). What makes one think that there might be a problem with autofluorescence is that the RFP+ cells seem to be phagocytotic bloated cells (Fig. 1). There don’t seem to be many monocytes (much smaller cells). Also CCR2 is usually very quickly downregulated when the monocte infiltrates the tissue, but here we find a lot of CCR2+ cells. So corroboration by another technique of this CCR2+ cell infiltration would be important.

Protective effect of Minocycline: The group has previously shown that CCL2 and CCL3 deletion very significantly preserves photoreceptors in 8 week old Mertk-/- mice. In their 2014 paper control mice were left with less than 15µm of ONL in Mertk-/- mice. Here at the same age we have more than 30µm in controls and the minocycline treated mice are not all that different. To be convincing I think a longer treatment and a later stage in the degeneration with a bigger difference would help…

Mechanism: finally it would be nice to have a little more insight in the mechanism. Does minocycline change the expression levels of CCL2 and CCL12, the two main ligands of CCR2 in mice? Does it change the populations of circulating monocytes? Or does it affect immunosupressivity in the eye? Some of these questions should be very easy to get answers to and would strengthen the paper.

6. PLOS authors have the option to publish the peer review history of their article (what does this mean?). If published, this will include your full peer review and any attached files.

Reviewer #1: No

Reviewer #2: No

---

## [Author Response · Author response to Decision Letter 0]

17 Mar 2021

Reviewer #1: 

1. Title: The current title is ambiguous and can benefit from rephrasing. Both “Ccr2 suppression” and “Cx3cr1/Ccr2-visualized” terms are imprecise and hard to understand. I suggest stating the main conclusions in the title: “Minocycline administration diminishes CCR2-positive monocyte number in the retina and ameliorates photoreceptor degeneration in a mouse model of retinitis pigmentosa”.

Thank you for the comment, and in accord with the reviewer’s advice, we had revised the title of the manuscript.

2. Abstract: The terms here and elsewhere relating to the suppression of Ccr2 expression are imprecise. What the authors document here is a decrease in the prevalence of CCR2-RFP+ cells in the outer retina and RPE layers with minocycline. Whether this finding resulted from the decreased recruitment of CCR2-expressing monocytes versus the actual suppression of CCR2 expression in monocytes resulting in a decrease in RFP-fluorescence is not investigated. The authors may want to reconsider the use of terms that indicate that CCR2 expression is decreased in individual monocytes from minocycline’s action (like “CCR2 suppression” ). The authors should pay attention to the CCR2 vs. Ccr2 case nomenclature throughout the manuscript to make sure the correct term is used for gene and protein names. The abstract can probably be shortened with the omission of methodological detail.

Thank you for the comment. We agree that the terms relating the suppression of Ccr2 expression were not appropriate since, as the reviewer pointed out, we did not investigate whether the decrease of RFP-fluorescence was resulted from the decreased recruitment of CCR2-expressing monocytes or the actual suppression of CCR2 expression in monocytes. We had changed the wording to “decrease of the number of CCR2-RFP+ cells”. We apologize the usage of terms for gene and protein names was not paid attention. We had revised the terms in the whole manuscript. We removed the sentences describing methodology from the abstract. 

3. Results: It is important for the authors to establish what is being referred to as Cx3cr1-positive, Ccr2-positive and Cx3cr1/Ccr2 dual-positive cells are indeed GFP+/RFP-, GFP-/RFP+, and GFP+/RFP+ cells respectively. I had been looking throughout the manuscript for a confirmatory statement of this but I was not able to find it. The confusion is increased in Fig 2 and 3 in which the numbers in A-4, B-4, and C-4 do not match corresponding numbers in F, G, and H; in F, G, and H; dual-positive cells constitute the smallest constituency of all 3 cellular groups for all 3 conditions but in the pie charts, they consistently outnumber CCR2-postive cells. Without harmonizing this confusion, it is not possible to fully understand the data. 

We apologize the confusing figures in Fig. 2 and Fig. 3. In the pie charts; A-4, B-4, and C-4 (in the original figure), “Cx3cr2” and “Ccr2” represent GFP and FRP single positive cells, respectively (Cx3cr2 = GFP+RFP- and Ccr2 = GFP-RFP+). On the other hand, the panels D and E (in revised figure) indicate cell number of the sum of single positive and double positive cells; Cx3cr1 represents the sum of GFP+RFP- and GFP+RFP+, and Ccr2 represents the sum of GFP-RFP+ and GFP+RFP+. We found that information of the pie charts and the bar graphs are redundant. Therefore, we had removed the pie chart from the revised version. Related sentences had been revised and labelling of the graph and legend to the Fig 2 and Fig. 3 had been modified accordingly. 

4. Results: The ameliorative effect of minocycline on ONL thickness is actually quite minor and localized, leading one to wonder if it can be validated if more experimental numbers are accumulated. As such, the reference to this difference as “rescue” may be wording it too strongly. For smaller effects like this, more evidence is often desired to strengthen the conclusion, these include: (a) a dose dependent effect (i.e. more amelioration with higher doses of minocycline, like 100mg), (b) independent method of ascertaining ONL thickness (e.g. with in vivo  OCT measurements), and (c) other measurements of PR death (e.g. TUNEL labeling of apoptotic cells, length of outer segments, counting cone density etc). If these are available, they can help strengthen the conclusion that minocycline confers protection in this model.

In accord with the reviewer’s suggestion, we had conducted additional experiments. We increased sample numbers and added samples treated with higher dose (100 mg/kg) of minocycline. In total, we examined 6 eyes for each condition. We found that the 50 mg/kg and 100 mg/kg minocycline treated mice showed thicker ONL than those of control mice; however the difference between 50 and 100 mg/kg doses was not statistically significant. We added results of 100 mg dose treated mice to Fig. 5A, B, C, and E, and related sentences had been modified. Regarding the wording “rescue”, we agree with the reviewer’s opinion that the wording is too strong, and revised the sentences. 

5. Discussion: The authors can focus more on interpreting the main findings of the manuscript, asking the questions of (a) whether minocycline inhibits infiltration of CCR2-expressing monocytes into the outer retina, or whether CCR2+ monocytes are converted into dual positive cells, to CX3CR1+, CCR2- cells, or even to CX3CR1-, CCR2- cells (which may render them undetectable in this assay) and (b) whether this effect of minocycline on CCR2-expressing monocytes is causally connected with the amelioration of PR degeneration, and if so, by what putative mechanisms. These discussion points follow directly from the main results and should constitute the main content of the discussion.

Thank you for the comments, and we appreciate the reviewer to list the points of discussion. We agree with the reviewer that the discussion part should focus more on interpreting the main findings of the manuscript, and revised the whole part of the discussion 

Reviewer #2: 

Comment #1

One incertainty is that they only use fluorescent markers to identify the cell types although subretinal macrophages (whatever type, microglial or monocyte derived) are known to become very autofluorescent as they phagocyte visual pigment rich outer segments of the photoreceptors. In any of these studies it would therefore be important that other techniques are used to corroborate the results using non-fluorescent techniques, such as substrate based immunohistochemistry (e.g. Fast red, if there is an antibody that differentiates GFP and RFP that should be easy), or in this case a difference in CCR2 expression by RT-PCR (or other Monocyte markers Ly6C etc…). 

In accord with the reviewer’s suggestion, we employed diaminobenzidine based detection protocol to detect CCR2-RFP positive cells by immunostaining. As expected, we observed stained cells in the subretinal space, supporting the idea that the signals observed by fluorescent were indeed RFP signals and not auto fluorescent. We added the results as Fig. 4. Related sentences had been revised. We also performed RT-qPCR; however, the results were unstable and did not give statistically significant results, therefore we did not include the RT-qPCR results.

Comment #2 Protective effect of Minocycline: The group has previously shown that CCL2 and CCL3 deletion very significantly preserves photoreceptors in 8 week old Mertk-/- mice. In their 2014 paper control mice were left with less than 15µm of ONL in Mertk-/- mice. Here at the same age we have more than 30µm in controls and the minocycline treated mice are not all that different. To be convincing I think a longer treatment and a later stage in the degeneration with a bigger difference would help…

We agree that there was difference in ONL thickness in 8-week retina between the current study and our previous study, and have currently no rational explanation regarding the discrepancy. In accord with the reviewer’s suggestion, we examined effects of longer treatment of minocycline in Mertk knockout mice. We used 8-weeks mice as starting point, and after 8 weeks, we harvested the mice. We observed effects of minocycline to keep thickness of the ONL in this condition, but the degree of the difference of thickness of ONL between control and minocycline treated mice was similar with that observed in short-term minocycline treatment (from 6 to 8 weeks of age). However, the results supported the idea that the longer and later treatment of minocycline is also effective to reduce the degeneration. We added these results into Fig. 6, and related sentences were modified and added. 

Comment #3 Mechanism: finally it would be nice to have a little more insight in the mechanism. Does minocycline change the expression levels of CCL2 and CCL12, the two main ligands of CCR2 in mice? Does it change the populations of circulating monocytes? Or does it affect immunosupressivity in the eye? Some of these questions should be very easy to get answers to and would strengthen the paper.

Thank you for the valuable comments, and we examined the expression of these genes including immunosupressive cytokine genes by RT-qPCR, but we did not observe significant changes by the treatment of minocycline. We also examined peripheral circulating monocyte by a flow cytometer, and the population of the monocyte was not changed by the minocycline. We added monocyte results as supplemental Fig. 2, and related sentences had been revised accordingly.

---

## [Decision Letter · Decision Letter 1]

1 Apr 2021

Minocycline decreases CCR2-positive monocytes in the retina and ameliorates photoreceptor degeneration in a mouse model of retinitis pigmentosa

PONE-D-20-26780R1

Dear Dr. Kohno,

We’re pleased to inform you that your manuscript has been judged scientifically suitable for publication and will be formally accepted for publication once it meets all outstanding technical requirements.

Kind regards,

Alfred S Lewin, Ph.D.

Section Editor

PLOS ONE

Additional Editor Comments (optional):

Reviewers' comments:

Reviewer's Responses to Questions

**Comments to the Author**

1. If the authors have adequately addressed your comments raised in a previous round of review and you feel that this manuscript is now acceptable for publication, you may indicate that here to bypass the “Comments to the Author” section, enter your conflict of interest statement in the “Confidential to Editor” section, and submit your "Accept" recommendation.

Reviewer #1: All comments have been addressed

Reviewer #2: All comments have been addressed

2. Is the manuscript technically sound, and do the data support the conclusions?

Reviewer #1: Yes

Reviewer #2: Yes

3. Has the statistical analysis been performed appropriately and rigorously? 

Reviewer #1: Yes

Reviewer #2: Yes

4. Have the authors made all data underlying the findings in their manuscript fully available?

Reviewer #1: Yes

Reviewer #2: Yes

5. Is the manuscript presented in an intelligible fashion and written in standard English?

Reviewer #1: Yes

Reviewer #2: Yes

6. Review Comments to the Author

Reviewer #1: The authors have comprehensively addressed each of my comments. I have no further suggestions for improvement.

Reviewer #2: (No Response)

7. PLOS authors have the option to publish the peer review history of their article (what does this mean?). If published, this will include your full peer review and any attached files.

Reviewer #1: No

Reviewer #2: **Yes: **Florian Sennlaub

---

## [Editor Report · Acceptance letter]

12 Apr 2021

PONE-D-20-26780R1 

Minocycline decreases CCR2-positive monocytes in the retina and ameliorates photoreceptor degeneration in a mouse model of retinitis pigmentosa 

Dear Dr. Kohno:

I'm pleased to inform you that your manuscript has been deemed suitable for publication in PLOS ONE. Congratulations! Your manuscript is now with our production department. 

Kind regards, 

on behalf of

Dr. Alfred S Lewin 

Section Editor

PLOS ONE